Application of plasma metabolome for monitoring the effect of rivaroxaban in patients with nonvalvular atrial fibrillation

Zhao Mindi 1
Liu Xiaoyan 2
Bu Xiaoxiao 1
Li Yao 1
Wang Meng 3
Zhang Bo 1
Sun Wei sunwei@ibms.pumc.edu.cn 2
Li Chuanbao lcb9731@163.com lichuanbao3662@bjhmoh.cn 1
1 Department of Laboratory Medicine, Beijing Hospital, National Center of Gerontology, Institute of Geriatric Medicine, Chinese Academy of Medical Sciences , Beijing , China
2 Core Facility of Instrument, Institute of Basic Medical Sciences, Chinese Academy of Medical Sciences, School of Basic Medicine, Peking Union Medical College , Beijing , China
3 Department of Clinical Laboratory, Baoding First Central Hospital , Baoding , Hebei , China
Sistla Srinivas
Electronic publication date: 2022 Aug 9
Publication date: 2022
Volume: 10
Electronic Location ID: e13853
Received 2022 Mar 29; Accepted 2022 Jul 16
Copyright: ©2022 Zhao et al.
Copyright year: 2022
Copyright holder: Zhao et al.
License: This is an open access article distributed under the terms of the Creative Commons Attribution License, which permits unrestricted use, distribution, reproduction and adaptation in any medium and for any purpose provided that it is properly attributed. For attribution, the original author(s), title, publication source (PeerJ) and either DOI or URL of the article must be cited.
License URL: https://creativecommons.org/licenses/by/4.0/

Keywords: Plasma metabolomics, Atrial fibrillation, Rivaroxaban

Funding: Beijing Hospital BJ-2019-136 This work was supported by Beijing Hospital (BJ-2019-136). The funders had no role in study design, data collection and analysis, decision to publish, or preparation of the manuscript.

==============================
Rivaroxaban, an oral factor Xa inhibitor, has been used to treating a series of thromboembolic disorders in clinical practice. Measurement of the anticoagulant effect of rivaroxaban is important to avoid serious bleeding events, thus ensuring the safety and efficacy of drug administration. Metabolomics could help to predict differences in the responses among patients by profiling metabolites in biosamples. In this study, plasma metabolomes before and 3 hours after rivaroxaban intake in 150 nonvalvular atrial fibrillation (NVAF) patients and 100 age/gender-matched controls were analyzed by liquid chromatography coupled with mass spectrometry (LC–MS/MS). When compared with controls, a total of thirteen plasma metabolites were differentially expressed in the NVAF patients. Pathway analysis revealed that purine and lipid metabolism were dysregulated. A panel of three metabolites (17a-ethynylestradiol, tryptophyl-glutamate and adenosine) showed good predictive ability to distinguish nonvalvular atrial fibrillation with an area under the receiver operating characteristic curve (AUC) of 1 for the discovery phase and 1 for validation. Under rivaroxaban treatment, a total of seven metabolites changed, the lipid and glycosylphosphatidylinositol biosynthesis pathways were altered and the panel consisting of avocadene, prenyl glucoside and phosphatidylethanolamine showed predictive ability with an AUC of 0.86 for the discovery dataset and 0.82 for the validation. The study showed that plasma metabolomic analyses hold the potential to differentiate nonvalvular atrial fibrillation and can help to monitor the effect of rivaroxaban anticoagulation.

Introduction

Rivaroxaban, a novel oral anticoagulant (NOAC), is a direct oral factor Xa inhibitor and is available for clinical use across a series of thromboembolic disorders. Rivaroxaban has been approved for the prevention of stroke in patients with nonvalvular atrial fibrillation (NVAF), to treat venous thromboembolism in patients undergoing orthopedic surgery and for treatment of deep vein thrombosis and pulmonary embolism (Antoniou, 2015). Compared with traditional anticoagulants, NOACs feature several obvious advantages, such as unrestricted food intake and nearly unaffected drug interactions (Levy et al., 2014). However, increasing evidence suggests that laboratory testing is essential and useful to evaluate the anticoagulant effects and to define therapeutic regimens (Weitz & Eikelboom, 2016); for example, NOACs are associated with a significantly high risk of gastrointestinal bleeding in elderly individuals (Sharma et al., 2015). The intervariability of plasma concentration is reported to be approximately 30–40% for rivaroxaban (Gustafson et al., 2019) and is even more pronounced in seniors and individuals who are overweight or those who have hepatic or renal failure or who are undergoing multidrug therapy (Lippi & Favaloro, 2015; Tripodi, 2016). Measurement of the anticoagulant effect of rivaroxaban is valuable for monitoring drug response and ensuring the efficacy and safety of the drug.

Routine coagulation screening assays, such as prothrombin time, activated partial thromboplastin time, and thrombin time, are considered insufficient to assess the degree of NOAC effects, as the screening results vary due to differences in the sensitivity of reagents to NOACs (Dunois, 2021). Measurement of the plasma concentration of rivaroxaban by liquid chromatography–mass spectrometry is considered the gold standard method of laboratory testing (Gosselin et al., 2018). However, it is not applicable for rapid testing in laboratories due to the limited throughput, the absence of standardization assays and the lack of an international standard for calibration (Dunois, 2021). In addition, studies have shown that some genetic or acquired factors can amplify or weaken the anticoagulant effect of rivaroxaban (Gosselin et al., 2018), so the concentration of rivaroxaban cannot reflect the coagulation status and hemostasis risk of patients after drug administration. Anti-factor Xa assays are currently preferable for measuring rivaroxaban in clinical situations; however, studies have highlighted limitations, such as the lower limit of quantitation being approximately 30 ng/mL (Gosselin et al., 2018). Therefore, new laboratory testing methods and reagents are still needed to monitor the effect of rivaroxaban.

Metabolomics is characterized by a simultaneous measurement of a large quantity of metabolites in a biological sample. In addition to diagnostic and prognostic biomarker discovery, metabolomics is also used for drug safety monitoring and toxicity analysis and highlights some potential directions in accommodating the highly variable responses of patients to various drugs (Wishart, 2016). Metabolomics also helps to predict differences in response among patients by measuring plasma metabolites. For example, Bawadikji et al. (2020) applied nuclear magnetic resonance imaging to analyze plasma metabolites in patients with stable and unstable international normalized ratios (INRs) and found that alpha and beta glucose are useful biomarkers in patients with unstable INRs. Deguchi et al. (2014) applied mass spectrometry-based untargeted metabolomics and suggested that plasma procoagulant lipid molecules and ethanolamides are changed under therapy with the anticoagulant warfarin. These studies showed that the plasma metabolome varies with changes in coagulation status induced by anticoagulants, but to our knowledge, there is no metabolomic analysis for monitoring rivaroxaban therapy.

In this study, to monitor drug responses and predict the potential side effects of rivaroxaban, plasma samples from 150 nonvalvular atrial fibrillation (NVAF) patients before and 3 h after rivaroxaban intake (peak maximum plasma concentrations (Mueck et al., 2008)) and 100 gender- and age-matched healthy volunteers were characterized by liquid chromatography coupled with mass spectrometry (LC–MS/MS). Multivariate statistical models and functional and biomarker analyses were used to discover and validate differential metabolites in NVAF at different rivaroxaban concentrations. The study will provide insights for clinical laboratory monitoring of patients receiving novel oral anticoagulants. Rivaroxaban is not only used in atrial fibrillation patients, so the plasma metabolome analysis in this study may also provide new insights for clinical monitoring of rivaroxaban treatment for other diseases.

Materials & Methods

Ethical statement

The research protocol for this study was approved by the ethics committee of Beijing Hospital (2020BJYYEC-003-01). A total of 150 patients diagnosed with NVAF and 100 healthy volunteers were recruited from January 2019 to December 2020 in Beijing Hospital. Prior to study enrollment, all participants were given a verbal explanation of the study, and each participant signed an informed consent document. All methods in this study were performed in accordance with the guidelines and regulations.

Sample preparation

The enrolled NVAF patients were evaluated by cardiologists in Beijing Hospital as suitable for treatment with rivaroxaban and did not have rheumatic valvular disease or prosthetic heart valves (Camm et al., 2012). Patients with atrial fibrillation that was reduced by some reversible factors, such as perioperative onset and hyperthyroidism, were also excluded.

The study contained two parts, one is the plasma metabolome analysis of control group and patients with atrial fibrillation (before rivaroxaban intake); another one is the plasma metabolome analysis of patients with atrial fibrillation before and 3 h after rivaroxaban intake. The plasma was collected from controls, patients with atrial fibrillation (before rivaroxaban intake) and the same patient with atrial fibrillation (after 3 h rivaroxaban intake). For the 150 NVAF patients, plasma samples were collected before and 3 h after rivaroxaban intake. In addition, 100 plasma samples from age- and gender-matched healthy volunteers were collected to study the differences in plasma metabolites between controls and NVAF patients. After collection, all plasma samples were centrifuged at 3,000× g for 8 min at 4 °C and then stored at −80 °C. The plasma anti-factor Xa (FXa) activity, activated partial thromboplastin time (APTT), thrombin time (TT), prothrombin time (PT)/international normalized ratio (INR) were measured by using the ACL TOP 700 platform assay (Werfen, Barcelona, Spain).

For the metabolome study, the plasma samples were prepared as previously described (Liu et al., 2020). Briefly, 150 µL of H2O was added to 50 µL of plasma sample to dilute the sample; then, 400 µL of acetonitrile was added to the mixed sample. After vortexing, the mixture was allowed to stand for 30 min at −20 °C and centrifuged at 3,000× g for 8 min at 4 °C to remove plasma proteins. The supernatant was lyophilized and reconstituted with 2% acetonitrile. To avoid the interference of low molecular weight proteins, the sample was separated by using 10 kDa molecular weight cutoff ultracentrifugation filters (Millipore Amicon Ultra, MA) before MS analysis.

The quality control (QC) sample was a pooled plasma sample prepared by mixing aliquots of two hundred samples across different groups. The samples were randomly selected from NVAF patients and control groups. To evaluate the stability and repeatability of the strategy, the QC sample was analyzed every ten samples throughout the analytical run.

LC–MS analysis

Each sample was analyzed on an HSS C18 column (3.0 × 100 mm, 1.7 µm) (Waters, Milford, MA, USA) by reversed-phase separation. The elution gradient was 2–100% buffer acetonitrile (flow rate = 0.5 mL/min) over 18 min. The column temperature was set at 50 °C. The eluted fractions were then analyzed with a Waters ACQUITY H-class liquid chromatograph coupled with an LTQ-Orbitrap Velos mass spectrometer (Thermo Fisher Scientific, MA, USA).

The full MS data were acquired from 100 to 1,000 m/z at a resolution of 60,000. The automatic gain control (AGC) target was set at 1 × 106, and the maximum injection time was set at 100 ms. For the MS/MS scan, the data were acquired at a resolution of 15,000 with an AGC target of 5 × 105, maximum injection time of 50 ms, and isolation window of 3 m/z. The optimal collision energy was set at 20, 40 and 60 for each target for higher-energy collisional dissociation (HCD) fragmentation.

Data processing

The raw data acquired by MS were processed by Progenesis QI 2.2 software (Waters, Milford, MA, USA) (Chen et al., 2008; Zhang et al., 2016). The peak alignment algorithm was carried out, and the peak intensity was normalized in Progenesis QI software. For metabolite identification, the MS/MS data were searched against the HMDB database (http://www.hmdb.ca/). The peak intensities were then analyzed using MetaboAnalyst 5.0 (Pang et al., 2021) (http://www.metaboanalyst.ca) to perform the missing value estimation, log transformation and Pareto scaling to make features comparable. Principal component analysis (PCA) and orthogonal partial least squares discriminant analysis (OPLS-DA) were conducted by using SIMCA 14.0 (Umetrics, Sweden) statistical software. To identify differential metabolites, the variable importance in projection (VIP) value indicating the influence of each term in the model was calculated in the OPLS-DA analysis. The false discovery rate (FDR) was used to assess the chance of false positives. Differential metabolites were defined as those with a VIP value > 1, a p value of protein expression < 0.05 and a fold change > 1.5 between the two groups.

Functional annotation

For candidate biomarker prediction and evaluation, the receiver operating characteristic (ROC) curve in MetaboAnalyst 5.0 (Pang et al., 2021) was launched. It also provides a complete and easily visualized sensitivity and specificity report. In this study, the random forest algorithm was chosen for ROC analysis. The differential metabolites were mapped to canonical pathways by quantitative enrichment analysis in MetaboAnalyst 5.0.

The detailed methods can be found in the Supplemental Information.

Results

The workflow of the untargeted metabolome study

To establish a comprehensive metabolite profile in response to rivaroxaban treatments, metabolites extracted from plasma were analyzed with untargeted LC–MS/MS. A total of 150 nonvalvular atrial fibrillation patients were recruited and randomly divided into a discovery group and a validation group by gender and age matching in the LC–MS analysis. To clarify the longitudinal plasma metabolome difference between the healthy and NVAF patients, a total of one hundred healthy age-, body mass index- and gender-matched volunteers were also characterized as the control group and divided into the discovery subgroup and validation subgroup. Table 1 shows the demographics of all participants. For the NVAF patients, the plasma before rivaroxaban intake was also used to compare with controls to find candidate biomarkers of NVAF. The QC samples showed a small variation (< ± 2SD) in the plasma metabolome analysis, consistent with the stability and repeatability of the strategy, as the tight clustering showed (Fig. S1). The LC files of representative metabolites identified were in Fig. S2.

Table 1 The clinical characteristics of patients with nonvalvular atrial filtration and healthy volunteers.

	Discovery group	Validation group	
	Atrial filtration	Control	Atrial filtration	Control	
	Before	3 h after		Before	3 h after		
Male	42	42	26	42	42	26	
Female	33	33	24	33	33	24	
Age	74.7 ± 11.2	74.7 ± 11.2	68.3 ± 12.9	73.2 ± 12.7	73.2 ± 12.7	66.7 ± 14.0	
Plasma samples	75	75	50	75	75	50	
anti-Xa(ng/mL)	0.2 ± 0.2	1.8 ± 0.8*		0.2 ± 0.2	1.8 ± 0.8*		
PT (s)	12.1 ± 1.2	18.9 ± 4.7*	11.7 ± 1.5	12.9 ± 1.7	19.9 ± 3.9*	11.6 ± 0.9	
INR	1.1 ± 0.1	1.6 ± 0.4*	1.1 ± 0.1	1.2 ± 0.2	1.8 ± 0.3*	1.1 ± 0.1	
APTT(s)	33.8 ± 3.9	42.8 ± 8.7*	32.6 ± 3.5	32.5 ± 3.9	43.5 ± 6.0*	32.8 ± 2.7	
TT (s)	12.4 ± 0.9	11.9 ± 2.9	12.7 ± 1.0	11.6 ± 1.1	11.8 ± 1.3	12.2 ± 0.9	
Notes.

* Compared with the NVAF (before rivaroxaban), the clinical metadata of the same patient (3 h after rivaroxaban intake) have a p value < 0.05.

The workflow of this study is shown in Fig. 1. First, LC–MS/MS-based plasma differential metabolome analysis was performed in 75 NVAF patients (before and 3 h after rivaroxaban intake) and 50 controls to discover candidate biomarkers for NVAF (before rivaroxaban intake vs. control), as well as for laboratory monitoring of rivaroxaban (before vs. 3 h after rivaroxaban intake). After passing strict screening criteria, the candidate biomarkers for distinguishing between NVAF patients and controls and differential metabolites for monitoring rivaroxaban were determined. Furthermore, the differential metabolites between the two groups were further externally validated by using an independent batch of the other 75 NVAF patients and 50 healthy controls.

Figure 1 Study workflow.

Distinguishing patients with NVAF (before rivaroxaban intake) from controls by the plasma metabolome

To identify the metabolomic differences between NVAFs and healthy controls, the plasma metabolites of NVAFs and controls were analyzed by untargeted LC–MS/MS. Then, the raw MS/MS data were searched against the HMDB database to identify metabolites. The quantitative metabolome data were then imported into SIMCA 14.0 software, analyzed by multivariate statistical models, and illustrated with plots. To distinguish patients with NVAF from controls, PCA and OPLS-DA statistical models were applied. The PCA score plot (R2X = 0.945, Q2 = 0.561) showed a significant difference between the controls and the NVAF patients (Fig. 2A). Then, the OPLS-DA strategy was also applied to yield a clear separation of these two groups (R2Y = 0.801, Q2 = 0.771, Fig. 2B). The permutation test was used to ensure model validity and predictability. The statistical analysis demonstrated that plasma metabolome could easily separate the NVAF patients from the controls, indicating that the plasma metabolome of NVAF patients differed significantly from that of control subjects.

Figure 2 Plasma metabolome analysis of patients with NVAF and controls.

(A) The PCA score plot distinguished NVAF from controls. (B) The OPLS-DA score plots showed significant differences between patients with NVAF and controls. (C) Relative intensities of differential metabolites in NVAF and controls. (D) Pathway analysis of the differential metabolites between the two groups. (E) A panel of three metabolites (17a-ethynylestradiol, tryptophyl-glutamate and adenosine) showed good predictive ability to separate nonvalvular atrial fibrillation from controls with an AUC of 1 for the discovery phase. (F) The AUC of the panel was 1 with an external validation group.

A heatmap (Fig. 2C) was developed to visualize and intuitively compare the plasma metabolite composition and intensity of the NVAF and healthy groups. Compared with controls, the NVAF patient plasma metabolites were enriched in protein catabolism and downregulated in lipid metabolism.

A total of thirteen plasma metabolites (p < 0.05, fold change > 1.5, and VIP value > 1) were differentially expressed, as detailed in Table 2. Pathway analysis of these metabolites revealed that the purine metabolic pathway, glycerophospholipid metabolism and glycerolipid metabolism were markedly dysregulated in the patients with NVAF (Fig. 2D). The AUC value of each further illustrates that these differential metabolites hold good diagnostic value for NVAF. Seven out of thirteen changed metabolites exhibited AUC values above 0.9, and the AUCs of the others exceeded 0.8. To better distinguish disease, a panel of three metabolites with AUC above 0.9 (17a-ethynylestradiol, tryptophyl-glutamate and adenosine) were used and showed excellent predictive ability with an AUC of 1 for the discovery phase (Fig. 2E; NVAF, n = 75; control, n = 50) and 1 for the external validation data (Fig. 2F; NVAF, n = 75; control, n = 50).

Table 2 The differential metabolites identified in patients with nonvalvular atrial fibrillation and controls.

ID	Name	Fold change	FDR	VIP	AUC	
HMDB0001926	17a-Ethynylestradiol	0.004826	8.03E−48	1.665963	0.99653	
HMDB0028723	Arginyl-Gamma-glutamate	0.098722	1.65E−30	1.501268	0.97573	
HMDB0013043	Prostaglandin PGE2 1-glyceryl ester	0.004147	5.58E−23	1.411605	0.96747	
HMDB0029082	Tryptophyl-Glutamate	0.13976	1.28E−21	1.36716	0.9748	
HMDB0007851	LysoPA(0:0/18:1(9Z))	0.06645	2.77E−20	1.356147	0.92347	
HMDB0000050	Adenosine	15.83	9.07E−21	1.321155	0.9488	
HMDB0005770	Tuftsin	0.16819	4.9E−16	1.273592	0.89653	
HMDB0028974	Methionyl-Hydroxyproline	0.20535	1.2E−15	1.225894	0.89147	
HMDB0034194	Jurubine	0.040397	5.38E−14	1.183557	0.87813	
HMDB0004710	9,10,13-TriHOME	2.3522	1.84E−14	1.127472	0.91227	
HMDB0009203	PE(18:4(6Z,9Z,12Z,15Z)/20:5(5Z,8Z,11Z,14Z,17Z))	2.1998	3.16E−14	1.121004	0.87227	
HMDB0029075	Threoninyl-Gamma-glutamate	0.007917	5.57E−11	1.101023	0.88387	
HMDB0011756	N-Acetylleucine	1.6202	2.41E−10	1.066537	0.82667	

Differences in plasma metabolome during rivaroxaban treatment

As mentioned above, the measurement of novel oral anticoagulants is important under certain circumstances to ensure the efficacy and safety of drug use. In this study, to provide clues for the laboratory assessment of rivaroxaban treatment, plasma metabolites before and 3 h after rivaroxaban intake were collected from NVAF patients and analyzed by LC–MS/MS. Comparative analyses were conducted to investigate the presence of plasma metabolites that were significantly altered in response to rivaroxaban treatment. The multivariate approach using a PCA plot revealed an apparent separation in metabolome before and 3 h after rivaroxaban treatment (R2X = 0.682, Q2 = 0.480, Fig. 3A). The distribution in the same individual was more dispersed 3 h after rivaroxaban intake. The multivariate method using supervised OPLS-DA models also demonstrated a marked difference between these two timepoints (R2Y = 0.479, Q2 = 0.384, Fig. 3B). Permutation tests were used to ensure the stability and validity of the supervised OPLS-DA models. The results described above show that plasma metabolites hold the potential to reflect changes during the process of rivaroxaban treatment in patients with NVAF.

Figure 3 Plasma metabolome analysis of before and 3 h after intake of rivaroxaban.

(A) The PCA score plot showed distinction between the two groups. (B) The OPLS-DA score plots showed significant differences between the two groups. (C) Relative intensities of differential metabolites of before and 3 h after intake of rivaroxaban. (D) Pathway analysis of the differential metabolites between the two groups. (E) A panel consisting of three metabolites, avocadene, prenyl glucoside and phosphatidylethanolamine, showed predictive ability with an AUC of 0.86 for the discovery dataset. (F) The AUC of the panel was 0.82 for the external validation group.

A heatmap representing metabolite abundances was generated to intuitively illustrate differences before and 3 h after rivaroxaban intake (Fig. 3C). Compared with the baseline, the similar functions did not cluster together. A total of seven metabolites that changed after rivaroxaban intake met the criteria of p < 0.05, fold change > 1.5 and VIP value > 1 (Table 3). When imported into the MetaboAnalyst website, the seven metabolites showed significantly distinctive roles. Pathway analysis of these changed metabolites revealed that glycerophospholipid metabolism and glycosylphosphatidylinositol (GPI) anchor biosynthesis were different before and 3 h after rivaroxaban intake (Fig. 3D). Three metabolites had an AUC above 0.8, along with four above 0.7. Then, multivariate AUC-based exploratory analysis was used to achieve better prediction and relatively stable performance. A panel consisting of three metabolites, namely, avocadene, prenyl glucoside and phosphatidylethanolamine (PE), showed the best predictive ability, with AUC values of 0.86 for the discovery dataset (Fig. 3E, n = 75) and 0.82 for the external validation dataset (Fig. 3F, n = 75).

Table 3 The differential metabolites identified during rivaroxaban treatment of patients with nonvalvular atrial fibrillation.

ID	Name	Fold change	FDR	VIP	AUC	
HMDB0031042	Avocadene	0.58456	4.68E−06	1.174606	0.73422	
HMDB33630	Lactapiperanol C	0.64858	5.89E−05	1.098904	0.70969	
	3,4,5-trihydroxy-6-{2-hydroxy-3-(4-hydroxy-3-methoxyphenyl)propanoyl}oxy}oxane-2-carboxylic acid	1.5116	2.14E−06	1.727014	0.78418	
HMDB0031876	Prenyl glucoside	1.5329	1.53E−06	1.902436	0.79609	
HMDB0009203	PE(18:4(6Z,9Z,12Z,15Z)/20:5(5Z,8Z,11Z,14Z,17Z))	2.9346	1.83E−08	1.970988	0.82613	
HMDB0029334	Nummularine B	3.3137	1.29E−08	1.96705	0.80569	
HMDB34503	2,3,23-Triacetylsericic acid	3.7038	1.29E−08	1.89881	0.81049	

Canonical correlation analysis of changed metabolites and clinical metadata

The APTT, TT, PT/ INR and anti-FXa of each sample were measured as indicated above. A canonical correlation analysis between rivaroxaban-related clinical coagulation indicators and rivaroxaban-related metabolites intensity was used to discover effective constituents. The cross-correlation heat map was used to visualize the relationship (Fig. 4). As shown, the changed metabolites (PE, prenyl glucoside, nummularine B, caffeoyl, carboxylic acid and triacetylsericic acid) have a highly positively correlated with anti-FXa (r > 0.5); while the avocadene, menthane, cortisol and lactapiperanol C have a highly negatively correlated with anti-FXa (r <  − 0.4). The positively correlated metabolites were upregulated after rivaroxaban intake, with elevated INR, anti-Xa and APTT values, as reported before (Douxfils et al., 2012). These may indicate that the plasma metabolome has the potential to reflect the coagulation status after rivaroxaban intake.

Figure 4 The canonical correlation analysis between clinical coagulation indicators and rivaroxaban-related metabolites intensity.

A cross-correlation heatmap of PT, INR, APTT, TT, anti-FXa and changed metabolites.

Discussion

In this study, the plasma metabolite profiles of healthy volunteers and patients with NVAF were analyzed, and the metabolome of NVAF patients before and 3 h after rivaroxaban intake were characterized. Certain plasma metabolites differed significantly in abundance between the NVAF patients and the controls, reflecting differences in canonical pathways, such as lipid metabolism and amino acid metabolism. Our results also suggest that plasma metabolomes can vary among patients during rivaroxaban treatment, which may provide clues for elucidating the mechanism of drug action and predicting medication side effects. This study provides the basic plasma metabolome characteristics of NVAF and rivaroxaban treatment. To increase the reliability of the data, the experimental group and the validation group were used to discover and confirm the differential metabolites, and both discovery group and confirmation group showed good discrimination and high AUC values.

Plasma metabolite regulation in nonvalvular-atrial fibrillation

Pathway analysis showed that the purine metabolic pathway, glycerophospholipid metabolism and glycerolipid metabolism were dysregulated in NVAF. We then compared the findings in this study with previous studies and found some common pathways involved in the regulation of atrial fibrillation. Previous untargeted proteomics, lipidomics and metabolomics studies have found enrichment of the purine and fatty acid metabolic pathways in atrial fibrillation by LC–MS (Zhou et al., 2019). Significant elevation of plasma uric acid, which is a product of purine metabolism in humans, has been observed in patients with atrial fibrillation (Tekin et al., 2013). As indicated above, purine metabolism is closely related to atrial fibrillation progression. Li et al. (2021) found downregulation of ApoA-I in patients with atrial fibrillation by isobaric tags for relative and absolute quantitation proteomics. Peroxisome proliferator-activated receptors (PPARs) activate this protein and regulate its participation in fatty acid degradation and the glycerophospholipid metabolic pathway (Tokutome et al., 2019). Glycerolipid metabolism has been found to be significantly enriched in some cardiovascular diseases such as hypertension and stroke through KEGG analysis of the genome-wide association studies catalog (Ji et al., 2017), which may provide clues for studying the role of this pathway in atrial fibrillation.

When compared with controls, 17a-ethynylestradiol, tryptophyl-glutamate and adenosine showed excellent predictive ability (Figs. 5A and 5B); thus, they may be potential biomarkers and may play important roles in disease progression. Among the three dysregulated plasma metabolites, adenosine has been thought to be closely associated with various cardiovascular diseases, including atrial fibrillation. It is widely believed that the release of adenosine is involved in the pathophysiology of atrial fibrillation (Guieu et al., 2020). The development of atrial fibrillation is related to increased levels of endogenous atrial plasma adenosine (Maille et al., 2019), high expression of adenosine A2A receptors in the left atrium (Hove-Madsen et al., 2006) and heterogeneous expression of A1R in the right atrium (Li, Hansen & Fedorov, 2016). 17a-Ethynylestradiol is an alkylated estradiol. Sex hormones may affect the risk of arrhythmias by changing cardiac electrophysiological parameters (Gillis, 2017). Wong et al. (2017) found a pathophysiological link between exogenous administration of estrogen alone as postmenopausal hormone therapy and increased atrial fibrillation risk in women. Thus, 17a-ethynylestradiol may be a valuable biomarker for atrial fibrillation occurrence and progression. Another dysregulated metabolite, tryptophyl-glutamate, belongs to the class of organic compounds named dipeptides and has been less studied to date and thus may warrant further exploration in the future.

Figure 5 Plasma metabolome provides valuable clues for the diagnosis of NVAF and monitoring rivaroxaban.

(A) The abundance distributions of the three metabolites 17a-ethynylestradiol, tryptophyl-glutamate and adenosine in the normal and NVAF groups. (B) The predicted AUC of the panel and the three separate metabolites. (C) The abundance distributions of the three metabolites avocadene, prenyl glucoside and phosphatidylethanolamine before and 3 h after rivaroxaban intake. (D) The predicted AUC of the panel and those of the three separate metabolites.

Plasma metabolite regulation during rivaroxaban treatment

Glycerophospholipid metabolism and glycosylphosphatidylinositol (GPI) anchor biosynthesis were found to be different during rivaroxaban treatment in patients with NVAF, indicating that these two pathways may play potential roles in the process of rivaroxaban treatment. There is currently no direct evidence that these two pathways are involved in the action of rivaroxaban. However, these two pathways have been reported to be associated with changes in coagulation. Glycerophospholipid metabolism is upregulated both with high concentrations of rivaroxaban and in NVAF patients. This metabolism plays a role in the biosynthesis of oxylipins and promotes inflammatory responses (Stephenson, Hoeferlin & Chalfant, 2017) and has been reported to be regulated by some traditional Chinese medicines that could promote blood circulation to remove blood stasis (Zhang et al., 2018). Glycosylphosphatidylinositol plays roles in linking cell membranes and functional proteins, and the biosynthetic pathway includes more than two dozen proteins encoded by the phosphatidyl inositol glycan (PIG) genes (Wu et al., 2020). The PIGA gene has been reported to be linked to paroxysmal nocturnal hemoglobinuria, a disease characterized by hemolysis and thrombosis (Takeda et al., 1993). Almeida et al. (2006) identified a PIGM gene mutation in a novel inherited disease characterized by venous thrombosis and seizures, suggesting that GPI biosynthesis is crucial for maintaining homeostasis of blood coagulation. These results suggest that these two metabolic pathways may be potential targets of the anticoagulant rivaroxaban.

For monitoring of rivaroxaban treatment, a panel of three metabolites, namely, avocadene, prenyl glucoside and PE, showed the best predictive ability (Figs. 5C and 5D). After taking rivaroxaban, plasma avocadene concentrations in patients with NVAF decreased. Avocadene is a fatty alcohol and has been shown to display a bacteriostatic effect as well as to exhibit anti-inflammatory activities (Lu et al., 2012). Since avocadene is one of the most abundant and active compounds in avocado seed extract (Pacheco et al., 2017), it is important to pay attention to whether it may modulate the effect of rivaroxaban when a patient eats similar foods. Another potential biomarker, plasma prenyl glucoside, was upregulated after rivaroxaban intake. Prenyl glucoside, which belongs to the class of glucosides, has been less studied. However, there have been many reports that other types of glycosides are closely related to coagulation. For example, rutin glycoside delayed prothrombin time and activated partial thromboplastin time in vitro (Choi, Park & Lee, 2021); quercetin glycosides decreased blood coagulation time and reduced the activity of clotting factors in neonatal asthmatic rats (Zhu et al., 2019). The effect of prenyl glucoside on blood coagulation needs to be further studied.

Elevated PE in NVAF (3 h after rivaroxaban intake)

Compared with levels in healthy controls, PE (18:4(6Z,9Z,12Z,15Z)/20:5(5Z,8Z,11Z,14Z,17Z)) was downregulated in the plasma of patients with NVAF (before rivaroxaban intake). The activity of PE was significantly enhanced 3 h after rivaroxaban intake. PE was the only lipid showing a significant changed pattern in the two comparisons. Phosphatidylethanolamine (PE) is one of the most abundant glycerophospholipids in eukaryotic cells and is associated with diverse cellular functions, such as membrane fusion and oxidative phosphorylation (Calzada, Onguka & Claypool, 2016). We compared the metabolic pathways obtained in this study with those previously reported and reached some consistent conclusions. Ruan et al. (2020) explored the role of dysregulated circular RNAs in human monocytes from patients with atrial fibrillation and found through GO analysis that PE acyl chain remodeling was one of the most enriched biological processes. Rivaroxaban is a direct oral factor Xa inhibitor. A previous study reported that PE inhibited blood coagulation activity in the factor Xa-prothrombin system but had no effects on the protein C/protein S reaction system in an in vitro reconstructed experiment (Tsuda, Yoshimura & Hamasaki, 2006). Factor X has also been found to exhibit stronger binding affinities in the presence of PE in an in vitro system, and the synergy of PE and phosphatidylserine enhanced the binding of all seven GLA domain-containing proteins and membranes in the coagulation cascade (Medfisch et al., 2020). Our metabolomic studies demonstrated significant alterations in PE in the plasma of rivaroxaban-affected patients, which is consistent with findings of some previous in vitro experiments showing that factor X has greater binding affinities in the presence of PE.

Limitations

To our knowledge, the study is the first to reveal plasma metabolome changes of nonvalvular atrial fibrillation with rivaroxaban treatment. It indicated that plasma metabolome has the ability to evaluate the changes in relation to nonvalvular atrial fibrillation and are available to monitor exposure of rivaroxaban. For the untargeted metabolomics study, the good reproducibility of QC sample showed that LC/MS/MS non-targeted platform was robust enough for serum metabolomics analysis. Moreover, the metabolite panel showed good differential results not in discovery samples, but in validation ones, which showed our results could reflect the metabolic change of exposure of rivaroxaban. Nevertheless, there are some limitations need to be considered. Firstly, it is a single center observational study, and the sample size is small. The population group under the study belongs to one country, the obtained results may not be globalized. A multicenter and large-scale study need to be conducted in the future. Secondly, the changed metabolites found in the preliminary study have to be validated in a larger cohort by a targeted approach with authentic standards. Thirdly, the study discovered the changed metabolites in nonvalvular atrial fibrillation patients and rivaroxaban measurement, but how these metabolites affect the patients are unclear. The relevant functional analysis by animal models should be investigated.

Conclusions

In this study, plasma metabolome analysis for monitoring rivaroxaban therapy in patients with nonvalvular atrial fibrillation was established from 150 patients. The results suggest that plasma metabolome holds the potential to reflect the changes accompanying atrial fibrillation and to provide new insight into its diagnosis and rivaroxaban medication monitoring. The candidate biomarker panel may facilitate the diagnosis of nonvalvular atrial fibrillation and laboratory monitoring of rivaroxaban.

Supplemental Information

Table S1 The identification and quantitation of identified metabolites in discovery and validation phase

Click here for additional data file.

File S1 Supplemental methods and figures

Click here for additional data file.

We thank all the members for their generous participation.

Additional Information and Declarations

Competing Interests

Author Contributions

Human Ethics

Data Availability

The authors declare there are no competing interests.

Mindi Zhao conceived and designed the experiments, performed the experiments, analyzed the data, prepared figures and/or tables, authored or reviewed drafts of the article, and approved the final draft.

Xiaoyan Liu conceived and designed the experiments, performed the experiments, analyzed the data, prepared figures and/or tables, authored or reviewed drafts of the article, and approved the final draft.

Xiaoxiao Bu performed the experiments, prepared figures and/or tables, and approved the final draft.

Yao Li performed the experiments, prepared figures and/or tables, and approved the final draft.

Meng Wang performed the experiments, prepared figures and/or tables, and approved the final draft.

Bo Zhang performed the experiments, prepared figures and/or tables, and approved the final draft.

Wei Sun conceived and designed the experiments, authored or reviewed drafts of the article, and approved the final draft.

Chuanbao Li conceived and designed the experiments, authored or reviewed drafts of the article, and approved the final draft.

The following information was supplied relating to ethical approvals (i.e., approving body and any reference numbers):

This study was approved by the Ethics Committee of Beijing Hospital (2020BJYYEC-003-01).

The following information was supplied regarding data availability:

The raw data are available at the iProX database: IPX0004204000.

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
