# Peer review of "Application of plasma metabolome for monitoring the effect of rivaroxaban in patients with nonvalvular atrial fibrillation"

_PeerJ, doi:10.7717/peerj.13853_

## Round 0.1 · original submission · Major Revisions

The reviewers have given their comments on the manuscript. After careful deliberations, we have to conclude with major revisions. Please modify the manuscript and submit it at the earliest.

Reviewer 1 ·

Basic reporting

In this paper, Zhao et.al. conduct plasma metabolomic analysis for monitoring of rivaroxaban therapy in patients with nonvalvular atrial fibrillation. They find a panel of metabolites that are changed in the control vs NVAF patients; and NVAF patients before and after rivaroxaban treatment. They additionally use this panel to validate remaining set of NVAF patients and controls.
Although they have a found many metabolites to be differentially regulated in both the comparisons, my main concern is they have not carried out experimental validation of the same in this study.

Additionally these are some of the minor concerns.
1. The ethical review document is in Chinese with no translated document provided.
2. The emails of the corresponding authors should be from an official account
3. The whole manuscript should be checked for grammatical errors and ease of understanding. Some examples are
Line 59: individuals who are overweight
Line 76 needs rewriting
In places wherein 'plasma metabolomics' is mentinaled should be changed to 'plasma metabolome'

Experimental design

These are my concerns in the experimental designs:

1. Line 114: Before collection, rivaroxaban was taken once orally for at least four days. This line is not understandable. It is not clear from the statement if the patients were given the drug before sample collection.

2. Line 183: It is not clear which groups are being compared, controls vs NVAF (before or after rivaroxaban)

3. Figure 2C and 3C: Authors should mark which metabolites belong to which pathway and whether the pathway metabolites cluster together

4. The authors should mention as to how were the patients divided into experimental and validation groups

5. Have the authors monitored rivaroxaban levels in the plasma

6. The figure legends description is incomplete. Please specify what is 1 and 2 in Fig 4A and C

Validity of the findings

These are my concerns for validations:

1. It would be helpful if the authors can carry out an analysis of the selected metabolites using the metadata from the patients and associate the two together.

2. Can the authors carry out validation experiments for some of the metabolites that were differentially regulated in the groups and exploring how they affect the patients.

Additional comments

No comment

Reviewer 2 ·

Basic reporting

1. Fig 1 workflow has to be improved to make it clear to the readers

Experimental design

Most of my comments are minor and require revision
1. Line 131: gradient is water or buffer; please clarify
2. During the metabolomics studies, did the researchers use internal standard? If not, explain why?
Since the plasma has interfering compounds, they may strengthen or reduce the peak or AUC detection, leading to erroneous results.
3. Please include the HPLC file in the supplement and peak extraction figs of all three significant metabolites.

Validity of the findings

The impact has to be included in the discussion section.

The authors have to include a small paragraph in the discussion, what are the Strengths (robustness of the method, analysis) and Limitations ( like the population group under study belongs to one particular country and other variables in the patient group, the obtained results may not be globalized) of this study.

Additional comments

This is a well-written and competently performed study by investigators that provides information about Plasma Metabolomics in patients with nonvalvular atrial fibrillation.

---

## Round 0.2 · accepted · Accept

Based on the reviewers assessment I am happy to accept your paper for publication

Reviewer 1 ·

Basic reporting

The authors have considerably revised the manuscript and I am happy with the revisions.

Experimental design

no comment

Validity of the findings

no comment

Reviewer 2 ·

Basic reporting

No Comments

Experimental design

No Comments

Validity of the findings

No Comments

Additional comments

The authors tried to address most of the comments raised by the reviewers. The manuscript could be accepted in the current form for publication.